# Vertebrate Animal Models of RP59: Current Status and Future Prospects

**DOI:** 10.3390/ijms232113324

**Published:** 2022-11-01

**Authors:** Steven J. Fliesler, Sriganesh Ramachandra Rao, Mai N. Nguyen, Mahmoud Tawfik KhalafAllah, Steven J. Pittler

**Affiliations:** 1Departments of Ophthalmology and Biochemistry, Neuroscience Graduate Program, Jacobs School of Medicine, State University of New York—University at Buffalo, Buffalo, NY 14203, USA; 2Research Service, VA Western NY Healthcare System, Buffalo, NY 14215, USA; 3Department of Optometry and Vision Science, Vision Science Research Center, School of Optometry, University of Alabama at Birmingham, Birmingham, AL 35294, USA

**Keywords:** *cis*-prenyltransferase, DHDDS, dolichol, nogo-B receptor, retinal degeneration, RP59

## Abstract

Retinitis pigmentosa-59 (RP59) is a rare, recessive form of RP, caused by mutations in the gene encoding DHDDS (dehydrodolichyl diphosphate synthase). DHDDS forms a heterotetrameric complex with Nogo-B receptor (NgBR; gene *NUS1*) to form a *cis*-prenyltransferase (CPT) enzyme complex, which is required for the synthesis of dolichol, which in turn is required for protein *N*-glycosylation as well as other glycosylation reactions in eukaryotic cells. Herein, we review the published phenotypic characteristics of RP59 models extant, with an emphasis on their ocular phenotypes, based primarily upon knock-in of known RP59-associated DHDDS mutations as well as cell type- and tissue-specific knockout of *DHDDS* alleles in mice. We also briefly review findings in RP59 patients with retinal disease and other patients with *DHDDS* mutations causing epilepsy and other neurologic disease. We discuss these findings in the context of addressing “knowledge gaps” in our current understanding of the underlying pathobiology mechanism of RP59, as well as their potential utility for developing therapeutic interventions to block the onset or to dampen the severity or progression of RP59.

## 1. Introduction

Retinitis pigmentosa (RP), rather than being a single disease entity, is a family of distinct hereditary retinal degenerative diseases that share some phenotypic features, including progressive and irreversible loss of peripheral vision, due to initial dysfunction and death of rod photoreceptors, followed eventually by cone photoreceptor dysfunction and death, ultimately resulting in complete blindness [1]. In 2011, a previously unknown, rare, recessive, non-syndromic form of RP was recognized, where the molecular defect involved mutations in the gene encoding the protein, DHDDS (dehydrodolichyl diphosphate synthase; OMIM# 60872) [2,3]. This form of RP was named “RP59” (OMIM# 613861) and exhibits most of the canonical features associated with RP.

While the frequency of RP59 is rare in the general population, it has been estimated RP59 accounts for nearly one-third of RP cases involving Ashkenazi Jewish patients. The most common *DHDDS* alleles (associated with visual deficits) reported to date are homozygous K42E (i.e., replacement of Lys with Glu at position 42 of the polypeptide chain) and compound heterozygous K42E/T206A (replacement of Thr with Ala at position 206); additional alleles (e.g., K42E/R98W and W64X/C148EfsX11 that exhibits ocular and systemic features) also have been reported [2,3,4,5,6,7,8,9,10]. DHDDS is the catalytic component of the *cis*-prenyltransferase (CPT) enzyme complex, that also includes Nogo-B receptor (NgBR; OMIM# 610463) as a partner subunit; this heterotetrameric enzyme complex is required for synthesis of the long-chain isoprenoid lipid-soluble molecule, dolichol (Dol), and its phosphorylated derivatives, e.g., dolichyl phosphate (Dol-P) [11,12,13,14]. The only established biological function of these isoprenoids is to serve as the obligate glycan carriers for protein *N*-glycosylation and *O*- and *C*-mannosylation as well as glycosylphosphatidylinositol (GPI) “anchor” synthesis [15,16,17,18,19].

The dolichol-dependent (“lipid-linked”) oligosaccharide pathway is present and active in the vertebrate retina [20,21,22]. Blocking Dol-P-dependent glycosylation with pharmacological agents, e.g., tunicamycin, has been shown to result in disruption of photoreceptor outer segment (OS) morphogenesis and retinal degeneration [23,24,25,26,27]. Because of these facts, as well as findings obtained with a *dhdds* knock-down zebrafish model (see below), RP59 has been classified as a “congenital disorder of glycosylation” (or CDG) [28,29,30,31]. However, the reported cases of RP59 have not presented evidence that DHDDS activity has been compromised by the mutations so far associated with that disease, nor have they directly demonstrated that RP59 entails defective glycosylation. Hence, it seems unwarranted (or at least premature) to classify RP59 as a CDG. Additionally, several recent reports have described *DHDDS* variants with a limited range of non-ocular pathological features, including progressive myoclonic epileptic seizures, tremor, hypertonia, myoclonic status epilepticus, and congenital malformations [32,33,34,35,36,37,38,39,40,41]. One of these reports identified a patient with a specific deletion of amino acid K42 without ocular abnormalities [35].

Curiously, despite the fact that every cell type and tissue in the body carries out dolichol-dependent glycosylation reactions and, in fact, require such capability for cellular viability and to achieve and maintain cellular differentiation, RP59 has been classified as “non-syndromic”, i.e., pathology is primarily manifested in the eye/retina, rather than involving other bodily organs and tissues [4,42]. The reason for this is unknown: for example, there are no known retina-specific functions of DHDDS or dolichols. Here, we briefly summarize the progress made to date in generating and characterizing animal models of RP59, and new insights derived therefrom regarding the possible pathobiological mechanism underlying RP59.

## 2. Zebrafish Model of RP59

Zebrafish offer a tractable experimental vertebrate animal, readily obtained and maintained in a laboratory setting, for modeling a variety of human diseases, including retinal degenerative diseases [43,44]. A zebrafish *dhdds* transient knockdown model of RP59 was generated, using morpholino oligonucleotides injected at the one-cell embryo stage [45]. By postnatal (PN) day four, *dhdds* knockdown zebrafish failed to respond to the offset of light (unlike uninjected or sham-injected zebrafish) and exhibited dramatically shortened or missing photoreceptor outer segments (OS) compared to wildtype age-matched controls. Using fluorescently tagged peanut agglutinin (PNA), which selectively binds to a disaccharide that is present in the extracellular matrix surrounding cone photoreceptors (called the “cone matrix sheath”, or CMS) [46,47], the retinas of *dhdds* knockdown fish exhibit little or no PNA binding, whereas control fish retinas’ CMS were robustly labeled. While the observed decrease in PNA binding has been interpreted as lack of glycosylation, it may be alternatively due to early loss of cone photoreceptors.

Based on these results, the authors concluded that suppression of functional DHDDS compromises the viability of cone photoreceptors and the formation and maintenance of the cone OS. They further posited that these findings “support the hypothesis that insufficient DHDDS function leads to retinal degeneration”. It should be noted that, unlike human or rodent retinas, the photoreceptor population in the zebrafish retina is cone-dominant (ca. 50–60%), as opposed to rod-dominant, and PNA does not bind to the extracellular matrix that surrounds the rod OS. Unfortunately, this putative RP59 animal model, to date, has not been more expansively characterized, and it is yet unclear how relevant this zebrafish model is to RP59.

## 3. Mouse Models of RP59

The mouse is the most commonly used vertebrate animal model in experimental biology, including in studies relevant to eye and vision research [48,49]. The mouse retina is a “duplex” retina (i.e., possesses both rod and cone photoreceptors), but the rods heavily dominate the photoreceptor population (ca. 97%), while the cones are far less numerous. Additionally, unlike the human or nonhuman primate retina, the mouse retina lacks a cone-rich macula or fovea. Nonetheless, the fundamental cell biology and physiological processes that are extant in the mouse and human retina are quite comparable. Further, DHDDS is largely conserved in mammals; specifically, murine and human DHDDS exhibit 92.4% sequence identity with a 100% query coverage (E = 0.0) (using BLASTP algorithm) [13,50,51]. Hence, there have been efforts to generate murine models of RP59, using both knock-in and knock-out genetic manipulation of the *Dhdds* gene.

### 3.1. K42E Dhdds Knock-In Mouse

Using CRISPR-Cas9 gene-editing technology, a viable mouse model of RP59 has been generated (on a C57BL/6J background), and the ocular phenotype of mice either heterozygous (*Dhdds*^K42E/+^) or homozygous (*Dhdds*^K42E/K42E^) for this mutation has been reported [52]. The morphological organization of the retina of *Dhdds* mutants and age-matched wildtype (WT) controls was evaluated in vivo, using spectral domain-optical coherence tomography (SD-OCT), a non-invasive and quantitative analytical method. Surprisingly, no obvious structural abnormalities were observed as a function of *Dhdds* mutation, at least up to PN 12 months. Quantification of the total thickness of the neural retina as well as the thickness of the outer nuclear layer (ONL), the latter being specifically relevant to the health and persistence of photoreceptors, was performed to assess retinal structure. No overt differences in these quantifiable parameters were observed, comparing mutant and age-matched wild type (WT) mice. Hence, unlike human RP59, this mouse model did not appear to exhibit obvious retinal degeneration, even up to one year of age.

Retinal frozen sections were probed with antibodies against glial fibrillary acidic protein (GFAP; a biomarker of astrocytes and glia) and opsin (the visual pigment apoprotein; a biomarker for rod photoreceptors, especially the OS) (Figure 1).

Despite the seemingly normal appearance of the retina, anti-GFAP immunoreactivity was markedly elevated in *Dhdds*^K42E/K42E^ mice, relative to controls, and the radial labeling pattern reaching from the internal limiting membrane (ILM, the vitreoretinal interface) to the ONL was indicative of gliotic reactivity. However, opsin immunolocalization was comparable in mutant and control retinas, and there was no evidence of mislocalization of opsin (i.e., opsin was almost exclusively localized to the rod OS, rather than being partially distributed along the plasma membrane of the rod inner segment or down the axonal process or to the synaptic ending, as is commonly observed when rods undergo degeneration). Additionally, concanavalin-A (Con-A) lectin cytochemistry, with and without PNGase-F treatment, was performed to assess protein glycosylation status of *Dhdds*^K42E/K42E^ and WT mice (Figure 2). [PNGase-F treatment was done to simulate the scenario of a glycosylation defect, and to demonstrate that the lectin cytochemistry approach could detect such a defect, if present, in the mutant retinas.] Con-A labeling and PNGase-F sensitivity were comparable in mutant and control retinas. Hence, there was no evidence of globally compromised protein *N*-glycosylation in homozygous K42E *Dhdds* mutant retinas.

That initial report did not include any electrophysiological (electroretinography, ERG) analysis. However, subsequently, it has been discovered that this knock-in mouse model of RP59 exhibits initially subtle and then progressively more marked ERG defects. Starting at about PN 1 month of age, there are progressive reductions in the dark-adapted (scotopic; rod-driven) and light-adapted (photopic; cone-driven) ERG b-wave amplitudes, while the a-wave amplitudes exhibit no significant reductions [53], i.e., a “negative b-wave”. Those findings suggest there is defective transmission of visual information from rod and cone photoreceptors to their respective bipolar cell populations. Additionally, a more recent study has further evaluated the presence of structural abnormalities in the retinal pigment epithelium (RPE) and in the inner retina of the homozygous K42E *Dhdds* mutant mouse at PN 18 months, compared to age-matched C57BL/6J (WT) mice [54]. Grossly, retinal histology was comparable in both the mutant and WT mice. However, ectopic rod photoreceptor nuclei were found intermittently in the OPL of mutant retinas, and second-order neuronal processes were reduced, especially in the periphery. Pyknotic nuclei also were observed in the outer and inner nuclear layers (ONL, INL), and TUNEL-positive cells (consistent with apoptotic cell death) were far more numerous in retinas of the mutant than in WT mice. Ultrastructural features of the OPL and INL were mostly comparable in both mutant and WT retinas, however, some photoreceptor cell bodies and their synaptic terminals displayed darkened cellular material (consistent with impending cell death). In addition, the RPE basal infoldings adjacent to Bruch’s membrane often were disorganized and, on occasion, moderately to severely degenerated RPE cells were observed in K42E mutant mice. Hence, despite the grossly normal appearance of the mutant retina, these results suggest that defective signal transmission between photoreceptors and inner retinal neurons, as well as RPE dysfunction and compromised viability may be significant contributors to the etiology of RP59.

In a preliminary study (L. Surmacz and E. Swiezewska, unpublished results), it was found that this global K42E *Dhdds* knock-in mouse is still competent to synthesize dolichols; however, the isoprenylog isoforms in retina, liver, and brain have shorter than normal chain lengths: Dol-17 predominates (Dol-17 >> Dol-18 >> Dol-19), whereas in controls the dominant species are Dol-18 and Dol-19. In fact, the total amount of dolichol in tissues from homozygous K42E mutant mice is *greater than* (rather than less than) normal. This finding of a shift to shorter chain length dolichol species in all tested tissues is consistent with what has been reported for human RP59 patient plasma and urinary dolichol profile, where the Dol-18/Dol-19 ratio from *DHDDS*^K42E/K42E^ and *DHDDS*^T206A/K42E^ patients is ca. 3, while for unaffected controls the ratio is ca. 1, and for heterozygotes it is ca. 1.5 [7]. In fact, the dolichol isoform profile now is recognized as a useful companion diagnostic tool for a range of CDGs [27]. However, analysis of dolichol chain length and total dolichol content has not been performed on tissue biopsies of RP59 patients, nor has it been demonstrated that RP59 patient tissues or bodily fluids exhibit decreased levels or loss of total dolichol content compared to unaffected human subjects. The role of shortened dolichol chain length and/or dolichol content in RP59-associated retinal degeneration remains to be investigated.

### 3.2. Rod-Specific Dhdds Knockout Mouse

Using Cre-lox technology, mice that express Cre recombinase under the control of the rod opsin promoter (Rho-iCre75; [55]) were mated with *Dhdds^flx/flx^* mice, harboring *loxP* sites flanking *Dhdds* exon 3, (both on a C57BL/6J background) to generate mice that had *Dhdds* ablated selectively in rod photoreceptor cells, starting at PN day 7 [56]. This approach more closely models the degenerative effects of RP59-associated severe mutations with expected *null* DHDDS activity (e.g., W64X) [6,13,14,56]. At PN 4 weeks of age (allowing sufficient time for complete maturation of the retina), the structure of retinas of *Dhdds* knockout mice was comparable to that of age-matched control mice (the latter being *Dhdds*^flx/flx^ iCre^−^ mice, rather than WT), as determined by SD-OCT (Figure 3) as well as correlative histological analysis; yet, there were subtle, but measurable, ERG deficits in the mutant retinas (predominantly in scotopic a-wave amplitudes), compared to age-matched controls.

By PN 5 weeks of age, however, about 50% of the photoreceptors had died and dropped out, rod OS lengths were dramatically reduced, and the ERG deficits (scotopic and photopic) were comparably profound. By PN 6 weeks of age, there were few if any remnant photoreceptor cells and ERG responses were nearly extinguished. Importantly, at PN 4 weeks (i.e., prior to any obvious histological defects in the retina were manifest), the total dolichol content of the neural retina was decreased by ca. 50% in mutant mice, compared to age-matched controls. Additionally, at this time point, the ONL in mutant mouse retinas was devoid of *Dhdds* mRNA, as detected by in situ hybridization, whereas mRNA content of the INL in both mutant and control retinas was comparable (serving as an internal control), thus validating the cell type-specificity and efficiency of the *Dhdds* ablation. In addition, at PN 5 weeks, the rod-specific *Dhdds* knockout retina exhibited marked gliotic reactivity, as evidenced by dramatically elevated GFAP levels, measured by Western blot analysis as well as immunofluorescence confocal microscopy. There also were signs of a localized neuroinflammatory process and phagoptosis, as Iba-positive cells (consistent with activated microglia) were observed to invade the outer retina, some of which were seen to engulf TUNEL-positive (dying/dead) photoreceptor nuclei, and the levels of ICAM (an inflammatory cytokine) increased by >5-fold in mutant retinas, relative to age-matched controls. In addition, as observed in the *Dhdds*^K42E/K42E^ mouse model, lectin cytochemical analysis revealed the lack of any obvious protein N-glycosylation defect in the retina of this rod-specific *Dhdds* knockout model. Additionally, Western blot analysis of mutant vs. control retinas showed no glycosylation defect either in rod opsin (the most prominent glycoprotein in the vertebrate retina [57]) or in LAMP2 (a glycoprotein biomarker for lysosomal membranes [58]).

In summary, selective ablation of *Dhdds* in rod photoreceptors results in a rapid, severe, and irreversible retinal degeneration, primarily involving the outer retina (photoreceptor layer). This would be expected to result in prevention of dolichol synthesis in rod photoreceptors and, in turn, a marked loss of dolichol content of the neural retina. Notably, this scenario is quite different from what occurs in the homozygous *Dhdds*^K42E/K42E^ mutant retina (*see above*).

### 3.3. RPE-Specific Dhdds Knockout Mouse

Using the same general strategy as employed to generate the rod-specific *Dhdds* knockout model, homozygous floxed *Dhdds* mice were mated with a Cre recombinase mouse line under the control of the VMD2 (vitelliform macular degeneration 2) promoter (both lines on a C57BL/6J background), to ablate *Dhdds* selectively in RPE cells [59]. Although the primary defect was localized to the RPE, there were concomitant morphological abnormalities evident in the neural retina as well, including a progressive retinal degeneration, cell loss, and thinning, apparent initially at about PN 1 month. By PN 3 months, pathological features were evident in the RPE and photoreceptor cells, although non-uniformly, across the retina. RPE cells were observed ectopically in the photoreceptor layer, there was patchy loss of photoreceptor cells, and the external limiting membrane (ELM) descended toward and abutted the RPE. Consistent with this marked retinal degeneration, progressively worsening scotopic and photopic ERG deficits were observed at PN 1, 2, and 3 months. Unexpectedly, however, electrophysiological defects also were observed (although substantially less severe) in heterozygous *Dhdds* mutants. [Note: typically, recessive diseases, by definition, do not manifest in heterozygotes.] This observation suggests the possibility of a functional change in the CPT enzyme complex that occurs prior to any obvious retinal structural changes and suggests that 50% CPT activity may be insufficient to fulfill its biological function in the RPE. Additionally, this finding predicts that carriers of *Dhdds* mutations also may develop visual dysfunction, depending on the nature of the mutation and other factors, such as genetic background and environment.

Given the multiple essential functions that the RPE plays in supporting the physiological health of the neural retina [60], it is not surprising that a primary molecular defect such as *Dhdds* ablation in the RPE would eventually result in compromising the health of the underlying neural retina. These results suggest that RPE dysfunction likely contributes significantly to the observed *DHDDS* mutation-initiated pathology in RP59, and that the underlying disease mechanism may transcend simple disruption of glycosylation.

### 3.4. Nogo-B Receptor Mutants as RP59 Models

Homozygous *NgBR* (*Nus1*) knockout in mice results in early embryonic (E6.5 or earlier) lethality, while heterozygotes are viable and do not exhibit a pathological phenotype [61]. Humans harboring a R290H *Nus1* mutation exhibit profound musculoskeletal and neurological pathology, as well as macular lesions and other visual system defects [61]; however, this disease is distinct from RP59. Additionally, fibroblasts from patients harboring the R290H *NUS1* mutation have been shown to have defective protein glycosylation [61]. To date, there have been no reports of *Nus1* mouse mutants as RP59 models.

### 3.5. Emerging New Mouse Models of RP59

Another DHDDS mutation identified in RP59 patients is T206A [4,5]. This has only been reported thus far as a compound heterozygous mutation with K42E in patients. A recent preliminary report [62] has described the generation and initial characterization of new RP59 mouse models, consisting of targeted knock-in of T206A/T206A (homozygous), T206A/+ (heterozygous), and compound heterozygous T206A/K42E *Dhdds* mutations. ONL thickness and total neural retinal thickness measurements were comparable to WT control values for all of these *Dhdds* mouse mutant lines (by SD-OCT). INL thickness in all homozygous mutants, however, were markedly reduced. ERG a-wave amplitudes (scotopic and photopic) were comparable to WT values, but the ERG b-/a- wave amplitude ratios at saturating flash intensities (scotopic and photopic) were significantly lower than WT values for T206A/T206A and T206A/K42E *Dhdds* mutants (i.e., a “negative b-wave” phenotype). Additionally, the photopic b-/a- wave ratio difference was greater for T206A/K42E than for T206A/T206A mutants. T206A/+ mutant amplitude ratio values were comparable to WT control values. These results are consistent with (and extend) those obtained with homozygous K42E/K42E *Dhdds* mutant mice (K42E INL measurements have not been reported) and implicate defective photoreceptor-to-bipolar cell synaptic transmission in these RP59 mouse models. They also suggest that the K42E mutation is more strongly pathological than is the T206A mutation. To date, a knock-in R98W *Dhdds* mutant mouse RP59 model has not been reported.

### 3.6. Retinal Degeneration in a Drosophila Dhdds Knockdown Model

Although not a vertebrate, it should be noted that a *Drosophila* model of RP59 has been generated and characterized [63]. *Drosophila* offers a tractable model system readily amenable to genetic manipulation, and it contains a genetic ortholog (*CG10778*) to the *DHDDS* gene. Targeted RNAi-mediated knockdown of this gene was embryonic lethal. However, targeted expression of *CG10778*-RNAi using the *glass multiple reporter* (GMR)-Gal4 driver (GMR-DHDDS-RNAi) in the eye disc and pupal retina at the larval stage caused an unusual retinal degeneration phenotype. Photoreceptors R2 and R5 exhibited a nearly normal rhabdomere structure (the invertebrate counterpart to the “outer segment” of vertebrate photoreceptors), yet exhibited cytopathological features in the nuclear region, whereas other photoreceptors exhibited retinal degeneration in all regions. Additionally, rhodopsin levels were dramatically reduced in mutant vs. wildtype flies, while there was massive amplification (accumulation) of endoplasmic reticulum (ER) membranes in the photoreceptors. These results indicate that the *CG10778* gene product is essential for the normal development of the *Drosophila* retina. By extension, despite the known, considerable differences in retinal architecture between flies and humans, the results suggest that DHDDS may be essential for the normal development of the vertebrate retina.

## 4. Discussion

The ocular phenotype of the *Dhdds* knock-in mouse models of RP59 generated to date is less robust than what has been observed in human RP59 patients. To date, ~167 patients with *DHDDS* mutations have been examined clinically. As shown in Table 1, considering the 38 patients where best corrected visual acuity (BCVA) was determined, a range of 20/20 to 20/400 was reported, indicating significant visual capacity remaining in all patients examined. While the degeneration is clearly more aggressive in patients, there are many common features in murine and human DHDDS-mediated disease. Both K42E/K42E and T206A/K42E affected patients and corresponding mouse knock-in models have phenotypes confined to the retina. Both show a shift to shorter chain dolichol species, there is no evidence for loss or reduction in total dolichol content, and no direct evidence for defective glycosylation. Both exhibit retinal dysfunction, but it is confined more to the inner retina in the mouse, rather the outer retina (as in humans). Considering the conservation of the dolichol pathway across vertebrate species and the other similarities, these knock-in mice offer suitable models of the human disease and should yield significant insight into the pathophysiological disease mechanism.

As the T206A mutation is only found heterozygously with the K42E mutation in patients, the phenotype contributed by the T206A allele could not be determined. The characterization of the homozygous T206A animal model shows that the phenotype is the same as observed in the homozygous K42E mouse but is slower to degenerate and not as robust. This finding suggests that the patients that are heterozygous for each allele may have a greater window of intervention and may be more easily treated.

In the knock-in mice, the retinal degeneration that is observed is mostly confined to the inner retina. Electrophysiological (ERG) deficits (notably, a “negative b-wave”, i.e., essentially normal a-wave amplitude, but reduced b-wave amplitude) are evident, as is marked gliotic reactivity. Additionally, there is no evidence that these mutations prevent the formation of dolichol, per se, and there is no evidence to suggest that these mutations result in defective protein glycosylation. The two cell type-targeted *Dhdds* knockout mouse models (RPE-specific and rod photoreceptor-specific) generated to date represent a “worst case scenario” of what severe variants of DHDDS-associated disease might look like (albeit that global *Dhdds* knockout would be lethal) [6,56,59]. Yet, there was no evidence of compromised protein glycosylation in those models, either [52,56]. Taken together, these findings bring into question the validity of categorizing RP59 as a CDG. 

Other than its role in protein glycosylation and GPI anchor synthesis, there are no established alternative biological functions of dolichol. Furthermore, other than the pathway that utilizes DHDDS (the CPT enzyme complex), there is no alternative biochemical pathway catalyzing the *cis*-condensation of isopentenyl pyrophosphate and farnesyl pyrophosphate, to ultimately synthesize dolichol. Since the K42E mutation does not prevent dolichol synthesis, but rather skews the overall population of dolichol isoforms to shorter than normal chain lengths, there is no a priori reason to presume this would perturb protein glycosylation. In yeast, it has been shown that dolichol chain length is dependent on energy and carbon source supply [64]. However, there are no published studies using mammalian cells, tissues, or organisms that provide a direct link between dolichol chain length and protein glycosylation efficiency, or that demonstrate an optimal chain length for supporting protein glycosylation. In fact, the lack of a frank glycosylation defect in the K42E mouse model, with altered dolichol chain length profile, strongly suggests that shorter dolichols are sufficient for normal oligosaccharide synthesis. In the case of the rod-specific *Dhdds* knockout, it is conceivable that there was a persistent remnant pool of dolichol species that was sufficient to support protein glycosylation prior to the onset of retinal degeneration. Recall that the opsin promoter (driving Cre recombinase) in that model would not have been active until PN day 7 in the rods, and the lectin binding and correlative Western blot analyses were performed about three weeks later.

It remains unclear why RP59 is “non-syndromic”, given the ubiquitous requirement for dolichol throughout the body. We, and another group [35], have speculated (without the encumbrance of data) that there may be one or more retina-specific binding partners that interact with DHDDS (other than NgBR) and whose interactions might be perturbed by *DHDDS* mutations. The recent report of a microdeletion that specifically removed codon 42 leading to brain neurologic disease, but not retinal disease, emphasizes the importance of the protein structure in the net effect of a *DHDDS* mutation. *DHDDS* mutations that cause seizures and other neurological brain deficits [32,33,34,35,36,37,38,39,40,41] have been reported as dominant gain-of-effect mutations. We hypothesize that these mutations are also due to true gain-of-function, involving unidentified brain protein (s) that abnormally interact with the mutant DHDDS protein. In a similar manner, the retina-specific *DHDDS* mutations are proposed to be due to a less common recessive gain-of-function that requires either no normal allele to compete or greater levels of expression of the mutant that are provided through two mutant alleles. The mutations leading to early mortality that have more global effects on systemic function may be due to actual effects on glycosylation. The validation of this hypothesis awaits determination of the underlying mechanism of the observed electrophysiological and other cellular hallmarks of the disease, which we expect will be provided through continued analysis of the current and emerging animal models of RP59.

There remains the possibility that there are different isoforms of the DHDDS gene/protein, which could be cell type- or tissue-specific. According to the UNIPROT database (https://www.uniprot.org/uniprotkb?query=DHDDS, accessed on 13 September 2022) and the UCSC genome browser (https://genome.ucsc.edu/, accessed on 13 September 2022), there may be four alternatively transcribed *DHDDS* isoforms. However, their differential expression or functions have yet to be demonstrated or defined in any tissue or organism.

One must consider the potential limitations of the currently available data and the utility of the genetically modified RP59 mouse models extant. First, these animal models have been generated only relatively recently and have yet to be fully characterized. However, the existence of multiple preliminary reports (e.g., conference abstracts) suggest that there will be a considerable amount of new information forthcoming in the near future. Second, while to date, none of these RP59 mouse models have exhibited overt protein glycosylation defects, one cannot rule out possible selective glycosylation defects on specific proteins (for example, see [65]). Third, RP59 patients harboring a homozygous K42E *DHDDS* mutation exhibit a retinal phenotype that predominantly affects the outer retina (photoreceptor layer and RPE cells). Yet, the corresponding knock-in mouse model does not reflect this same phenotype. The reasons for this difference remain unclear at present, but may be due, in part, to fundamental biological or lifespan differences between mice and humans. Fourth, as indicated above, the mouse retina has distinct anatomical differences from the human retina (e.g., lack of a macula or fovea), which may influence gene expression and gene network interactions. Some of those issues may be further evaluated by using a spatial transcriptomics approach [66,67]. Fifth, all knock-out models based upon the use of allegedly cell type-specific Cre recombinase-expressing mouse lines are only as valid as the rigor by which the lack of “leakiness” of Cre expression has been demonstrated [68,69]. Sixth, such knock-out models are not strictly analogous to the human disease, since there are no live RP59 patients harboring *DHDDS* mutations. As mentioned above, they represent a “worst-case scenario” that illustrates the essential nature of functional DHDDS. Finally, to date, none of the mouse models have shed light on the reason why RP59 is non-syndromic. Despite these limitations, and with the realization that this area of research is still in its infancy, these vertebrate animal models of RP59 have provided some foundational insights into the disease. In addition, they offer novel models for testing new therapeutic interventions to reduce the severity, delay the onset of, or even prevent RP59.

## Figures and Tables

**Figure 1 ijms-23-13324-f001:**
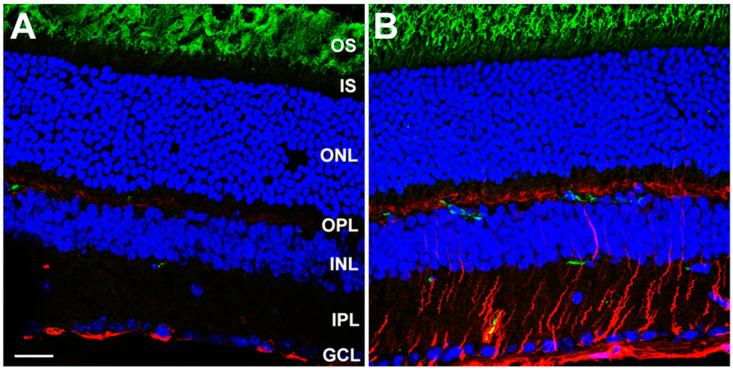
Immunofluorescence confocal microscopy images of (**A**) WT control and (**B**) *Dhdds*^K42E/K42E^ mouse retinas (age: PN 2 months). Binding of antibodies against GFAP (*red*) and rod opsin (*green*) is shown, with DAPI (*blue*) counterstain. Abbreviations: OS, outer segment layer; IS, inner segment layer; ONL, outer nuclear layer; OPL, outer plexiform layer; INL, inner nuclear layer; IPL, inner plexiform layer; GCL, ganglion cell layer. Scale bar (both panels): 20 μm. (Reproduced (open access) from Ramachandra Rao S et al., *Cells*, 2020, 9, 896 [52]).

**Figure 2 ijms-23-13324-f002:**
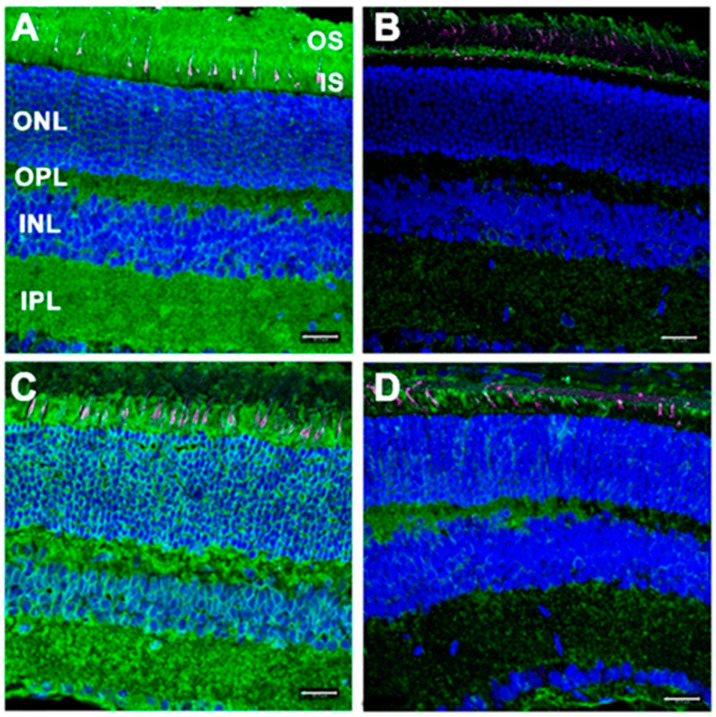
Binding of fluor-labeled Con-A (*green*) to retinas from (**A**,**B**) WT control and (**C**,**D**) *Dhdds*^K42E/K42E^ mice, with (**B**,**D**) or without (**A**,**C**) pretreatment with PNGase-F (age: PN 6 months). Binging of fluor-labeled peanut agglutinin (PNA; *magenta*) also shown; DAPI (*blue*) counterstain. Abbreviations: See legend, Figure 1. Scale bar (all panels): 20 μm. (Reproduced (open access) from Ramachandra Rao S et al., *Cells*, 2020, 9, 896 [52]).

**Figure 3 ijms-23-13324-f003:**
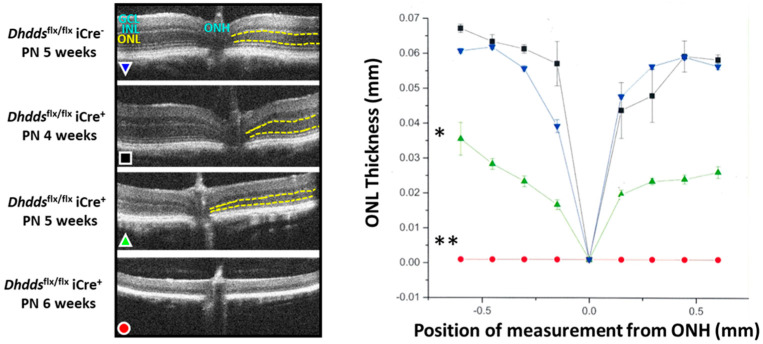
(**Left panel**) SD-OCT images of retinas from rod-specific *Dhdds* knockout (*Dhdds*^flx/flx^ iCre^+^) mice, compared to age-matched controls (*Dhdds*^flx/flx^ iCre^−^), between PN 4 and 6 weeks (color key included in panel). The ONL is demarcated by yellow dotted lines. (**Right panel**) Corresponding graph of ONL thickness measurements (in mm; *n* = 4 per genotype/age). * *p* < 0.05, ** *p* < 0.01; Welch’s (unpaired) *t*-test. Abbreviations as in legend, Figure 1; ONH, optic nerve head. (Reproduced, with permission, from Ramachandra Rao S et al., *iScience*, 2020, 23, 101198 [56]).

**Table 1 ijms-23-13324-t001:** DHDDS mutations and disease phenotype.

**Biswas [5]**	**Kimchi [4]**	**Hariri [10]**	**Reference**
*DHDDS* (p. K42E and p. T206A)	*DHDDS* (p. K42E, p. T206A and p. R98W)	*DHDDS* (p. K42E)	**Mutation**
4 (4)	30, 28 with K42E (22)	4 (4)	**N (*n*)**
AJ	AJ	NR	**Ancestry**
Negative FH	NR	NR	**Family History**
Night blindness started at age 17 y/o in one patient	Major loss of vision during the third and fourth decades	Mean age at diagnosis was 20 y/o (r, 17–22)	**Medical/Ocular History**
At age 22 y/o, 20/25 OU; at age 32 y/o, 20/40; at age 33 y/o, 20/50; at age 40 y/o, 20/70	NR	Four patients: 20/20 OU; (20/30 OD, 20/40 OS); (20/40 OD, 20/50 OS); (20/60 OD, 20/100 OS)	**BCVA**
Lens, PSC; Fundus, ONH pallor, attenuated retinal blood vessels, pigmentary RPE changes with white spots in the periphery; FAF, NR	Lens, NR; Fundus, atrophy and bone spicule pigmentation which increased in density and involved the macula with age; FAF, perifoveal ring of hyper-autofluorescence	Lens, NR; Fundus, NR; FAF, reduced AF of different grades, abnormal autofluorescence in the macula, complete disc hyper-autofluorescence in two patients	**Lens, Fundus, FAF**
Progressive constriction over time to <20° diameter by age 31 y/o	NR	NR	**Visual Field (VF)**
Scotopic and photopic ERG, severely reduced responses with similar reduction in a and b waves’ amplitudes	Scotopic ERG was non-detectable at first testing; Cone flicker ERG became non-detectable by 28 y/0	NR	**ERG**
NR	Loss/disruption of the ellipsoid zone, ONL, and RPE; small foveal islands of PRs which ultimately disappeared with age	NR	**OCT**
Chorioretinal biopsy, intact RPE with significant degeneration of all other retinal layers including GCL and PRs (inner and outer segments, and nuclei); Audiogram, bilateral normal hearing function	Color vision, tritanopia in two patients; EOG Arden ratio *** was reduced (r, 100–144%)	NR	**Others**
**Venturini [9]**		**Lam [8]**	**Reference**
*DHDDS* (p. K42E)		*DHDDS* (p. K42E)	**Mutation**
6 ** (5)		3 (3)	**N (*n*)**
Three of Jewish ancestry, two of mixed ethnicities		AJ	**Ancestry**
Positive FH		NR	**Family History**
Night blindness started at 27.8 y/o (r, 21–32)		Night blindness and peripheral vision defects by 15 y/o in two siblings	**Medical/Ocular History**
Five patients: 20/20 OU; 20/25 OU; 20/30 OU; (20/50 OD, 20/30 OS); (20/100 OD, 20/30 OS)		At diagnosis, from 20/20 to 20/25; in mid-thirties, from 20/40 to 20/400; one patient was LP OU by 30 y/o	**BCVA**
Lens, PSC in all (one case was pseudophakic OU); Fundus, peripheral bone spicule pigmentation in all, granular macula in 2 patients; FAF, NR		Lens, NR; Fundus, pigmentary retinal degeneration; FAF, NR	**Lens, Fundus, FAF**
VF loss onset at a mean age of 28.6 y/o; in fourth decade, most VF areas were below 50% of normal		Constricted to <10° at age 36 y/o in two siblings	**Visual Field (VF)**
Mean amplitude for Scotopic ERG was 21.9 mv (r, 2.6–47) and 19.6 mv (r, 1.4–44.1) for OD and OS, respectively; for photopic ERG was 2.1 mv (r, 0.21–5.9) and 2.01 mv (r, 0.25–5.64) for OD and OS, respectively		Non-detectable in two siblings	**ERG**
NR		NR	**OCT**
Dark adaptation, threshold in two patients was 0.5 and 1.5 log units above normal		Plasma transferrin isoelectric focusing gel, all patterns were normal; protein glycosylation was normal	**Others**
**Zelinger [3]**		**Zuchner [2]**	**Reference**
*DHDDS* (p. K42E)		*DHDDS* (p. K42E)	**Mutation**
21 (18, one family with longitudinal data)		3 (3)	**N (*n*)**
AJ		AJ	**Ancestry**
NR		NR	**Family History**
NR		Lytic bone disease in two siblings; retinitis pigmentosa diagnosis in teenage years	**Medical/Ocular History**
Ranged from LP to 20/20 (only four eyes with 20/20); 20/200 or worse in two siblings from the family with longitudinal data by age 30–31 y/o		NR	**BCVA**
Lens, NR; Fundus, waxy ONH; attenuated retinal blood vessels; bone spicule-like pigmentation; FAF, preserved RPE islands corresponding to regions of preserved PRs		Lens, NR; Fundus, pigmentary retinal degeneration; FAF, NR	**Lens, Fundus, FAF**
Reduced peripheral function; small central islands of vision remaining later in life		NR	**Visual Field (VF)**
Non-detectable in most patients; borderline rod ERG amplitude in one father’s recording from the family with longitudinal data		Impaired rod and cone responses	**ERG**
Preserved PRs’ layer in the fovea which declined in thickness away from the fovea; occasional CME; in the family with longitudinal data, PRs were not detectable around the fovea in two siblings, while one sibling had a locus of PRs nasal to the ONH		NR	**OCT**
Dark Adaptation, progressively diminished until only cone-mediated function was detectable; DHDDS staining, prominent in the basal aspect of RPE cells, IS of the cones, ellipsoid and myoid regions of the rods, weak signal in other retinal layers		Neurologic examination, bone X-ray survey and density scan, brain MRI, echocardiogram, lipid profile, thyroid function studies, serum IGF-binding protein 1 and 2, serum clotting factors, and antithrombin III were normal	**Others**

Abbreviations: AJ, Ashkenazi Jewish ethnicity; EOG, electrooculagram; ERG, electroretinogram; FAF, fundus autofluorescence; FH, family history; LP, light perception; NR, not reported; OD, right eye; OS, left eye; OU, both eyes; y/o, years old; PRs, photoreceptors; PSC, posterior subcapsular cataract. All other abbreviations as defined in the text. Lam 2014 reported additional data for the same family in Zuchner 2011. Autosomal recessive retinitis pigmentosa Patients with *DHDDS* mutations (*n* = 9), All AJ; Carriers of *DHDDS* mutations (*n* = 35), All AJ; Autosomal recessive retinitis pigmentosa Patients with WT-*DHDDS* (*n* = 34), One AJ; Normal Individuals (*n* = 19), 8 AJ. ** K42E mutation was homozygous in five patients and heterozygote in one patient. *** Arden ratio is the light peak/dark trough ratio in percent (normal > 185%).

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
