# Peer review of "Vertebrate Animal Models of RP59: Current Status and Future Prospects"

_ijms, 2022, doi:10.3390/ijms232113324_

Round 1

Reviewer 1 Report

This is an informative well-written manuscript worthy of review for the molecular mechanism associated with DHDDS arRP. Published “as-is”, this paper would be a nice manuscript but I have a question.

 When viewing Figs. 1 and 2, I noticed differences in the WT and mutant inner segments. However, upon reading the text, the authors report no differences. Am I seeing artifact in both images or could this be evident of impending pathology? Also, consider orienting the retinas in the same direction for Figs. 1 & 2.

Author Response

We thank this Reviewer for the kind and positive comments with recommendation to publish our manuscript.  With regard to the queries/comments:  “ When viewing Figs. 1 and 2, I noticed differences in the WT and mutant inner segments. However, upon reading the text, the authors report no differences. Am I seeing artifact in both images or could this be evident of impending pathology? Also, consider orienting the retinas in the same direction for Figs. 1 & 2.”

RESPONSE:   Figs. 1 and 2 are images obtained using frozen sections.  It is not possible to make definitive statements about morphological or ultrastructural elements at this level of resolution from such sections.  As stated, we found no appreciable differences (even in photoreceptor inner segments); at worst, any differences noted by the reviewer may be due to processing artifacts. With regard to orientation of the images-  Figs. 1 and 2 are shown in the same orientation (with photoreceptor layer at top and the ganglion cell layer at bottom).  Fig. 3 (left side, SD-OCT) is “flipped” in relation to what is shown in Figs. 1 and 2, as is the convention used in the field for such images.  At any rate, the histological layers are labeled to inform the reader of orientation.  Also, it should be noted that these images are reproduced from prior publications, to illustrate the key points of information discussed in the manuscript; hence, they retain the content and orientation as originally published.

Reviewer 2 Report

The aim of the paper from Fliesler et al. was to review the phenotypic characteristics of animal models based upon knock-in of known RP59-associated DHDDS mutations as well as cell type and tissue-specific knockout of DHDDS alleles in mice) of a  rare, recessive form of Retinitis pigmentosa, the so-called RP59, caused by mutations in the gene encoding DHDDS (dehydrodolichyl diphosphate synthase), which in complex with Nogo-B Receptor possesses a cis-prenyltransferase (CPT) activity, required for synthesis of dolichol, that plays a pivotal role in protein N-glycosylation. The paper also reviewed the findings in RP59 patients displaying retinal disease and those patients with DHDDS mutations displaying epilepsy and other neurologic impairments.  

The scientific question is original and well defined. Methods are correct and described with sufficient detail. The results are interpreted appropriately, and conclusions are justified and supported by the results. The paper is written in an appropriate way. and the data are presented appropriately.

A weakness of the paper is the interpretation of the data, that does not attempt to explain the ultimate reason for the almost exclusive involvement of the outer retina in the damage  in relation to relevant literature data. The conclusions were not adequately discussed.

It has been supposed that there exists a common pool of FPP in the cytosol , available for all the branch-point enzymes, depending on the different affinities of the branch-point enzymes, one of which is  cis-Prenyltransferase mediating the sequential cis-addition of IPP units, to all-trans-FPP ( see Grünler, et al.” Biochimica et Biophysica Acta - Lipids and Lipid Metabolism 1212.3 (1994): 259–277).

The deficiency of an enzyme leads to the accumulation of substrates upstream: if these are in a branched pathway, the other part of the pathway will be activated by the increased availability of the substrate, with a downstream increase in the end product, which in this case is ubidecarenone. In fact, one cannot only consider the dolichol deficiency as the cause of the damage, but one must also consider the consequence of the accumulation of FPP.Of course the diminished cis-Prenyltransferase DHDDS activity would increment the flow of precursors to the trans-prenyl transnferase, with a marked increase in CoQ synthesis. This in turn would cause an increase in the work of the respiratory chain, in fact it was shown that an increase in the mitochondrial Q-pool produces an increase in ROS generation doi: 10.1038/s41598-021-84852-z.) and that  the increase in CoQ affects  the mitochondrial membrane potential, and the Reactive Oxygen Species  production  https://doi.org/10.1016/j.bbabio.2004.10.009.

Considering that an ectopic respiratory chain and ecto-ATP synthase  have been shown to be expressed and catalytically active in the rod outer segments, this would explain why in fact  there are in fact the main targets of this gene deficiency. It seems to me that the authors should consider this explanation, cite the data and discuss it. if it discussed the data in light of the new findings of a source of oxidative stress inside the photoreceptor outer segments, besides the retinal mitochondria. The mitochondrial redox chain proteins as well as ATP synthase were shown to be ectopically expressed, not only in the mitochondria. In particular tis was found in the rod Outer Segments (see doi: 10.1096/fba.2019-00093.). Moreover, the involvement of an ectopic oxidative phosphorylation present in the myelin sheath would also represent an intriguing explanation for the forms involving epilepsy and other neurologic diseases, as a consequence of the DHDDS mutations.( doi: 10.1002/jnr.24865). 

Author Response

We thank this Reviewer for the very detailed critique and for providing additional points of information, some of which we had not originally considered in the course of writing our manuscript.

RESPONSE:  There clearly are differences between the biology of phylogenetically “lower” vertebrates (e.g., mouse) and humans, some of which likely are impacted by differences in life span (e.g., only a few years for mouse, vs. 7+ decades for humans). Another major difference between primates and other mammals (such as rodents) is that only primates have a macula. RP59 clearly has macular involvement, which may significantly contribute to the disease pathology.  So, the extant vertebrate RP59 models (we focus on genetically modifiable mice) admittedly do not faithfully recapitulate all of the pathological elements of the human disease.  We now expand on this in the Discussion, re: limitations of the current models.  The Reviewer should appreciate that RP59 was only recognized as a form of RP a little more than one decade ago and, probably due to the fact that it is a rare (“orphan”) disease, the relevant literature is somewhat sparse and a detailed account of the underlying cause(s) of the disease, beyond noting certain associated mutations in DHDDS, has yet to emerge.  At present, we have no compelling explanation (other than presence/absence of a macula) for why the K42E Dhdds knock-in mouse model does not exhibit a marked outer retina degeneration as expected.

  • While we greatly appreciate the very detailed (and erudite) commentary of this Reviewer, we think there has been a fundamental misunderstanding of what is going on in this disease. There is no evidence to date that the known DHDDS mutations in humans resulting in RP59 lead to “enzyme deficiency”, nor that there is a resultant dolichol deficiency.  To the contrary, the data clearly demonstrate that dolichol synthesis occurs, but there is a shift to shorter chain lengths (see Wen et al., 2013, cited in our manuscript).  Wen et al. did not report reduction in total dolichol levels in RP59 patient samples (plasma or urine) relative to unaffected controls, nor has anyone demonstrated shift to shorter chain lengths or reduction in total dolichols in any tissue (g., retina, liver, brain) in specimens from RP59 patients.  The only time we’ve seen a reduction in tissue dolichol levels has been when we ablated the DHDDS gene (see Ramachandra Rao et al., iScience 2020 Jun 26; 23(6):101198), as would be expected. Null mutations in DHDDS in humans (functionally equivalent to gene ablation) have not been observed in humans (in fact, they likely would be lethal).  As we discussed, we have new data (unpublished; currently under review by another journal), using the K42E Dhdds knock-in mouse model, that confirms the shift to shorter dolichol chain lengths in retina, liver and brain; yet, there is no loss or reduction in the levels of total dolichol content.  As we described in our review and in the original report we published on the K42E mouse model (Ramachandra Rao et al., Cells 2020 Apr 7;9(4):896), there was no detectable resultant protein glycosylation defect observed in the retina.  NOTE:  To our knowledge, there is no published work that has demonstrated a preferred chain length of dolichol species required for dolichol-linked oligosaccharide formation or transfer of that dolichol-linked glycan chain to protein by oligosaccharyltransferase.  Hence, despite the dominance of shorter than normal dolichol chain lengths as observed in specimens from RP59 patients (and in the K42E knock-in mouse model), there is no a priori reason to expect that this would translate to defective protein glycosylation, as would be found in a bona fide CDG (congenital disorder of glycosylation).
  •  
  • We found the remainder of this Reviewer’s commentary to be interesting, but consider it to be quite speculative--  and, there is good reason to believe it is unlikely.  First, the premise that DHDDS mutations as found in RP59 result in loss or appreciable reduction in Dol levels is not supported by the facts (see above); so, all of the speculation about the pathway backing up, causing changes in FPP or IPP or CoQ levels is rendered moot.  Second, in fact, there is strong evidence to the contrary. It has been demonstrated (see Keller RK et al. JBC 1988 PMID: 3257490) that the rate of C2 flux into the squalene-sterol branch in retina was 3.4 pmol/h/retina, while the Dol branch was 0.022 pmol/h/retina (a 154-fold difference). Also, in liver, the flux of carbon intermediates down the mevalonate pathway into Dol under normal conditions (but varies by dietary and diurnal conditions) may be ~500- to 1000-fold lower than what becomes incorporated into cholesterol (see Table III in Keller RK, JBC, 1986; PMID: 3638306).  So even if CPT activity was defective in RP59 (which it is not), any slight loss of FPP and IPP fluxing through the Dol pathway would have a negligible effect on FPP levels and downstream isoprenoid products. [It has been shown, however, that the reverse is true when one blocks just the sterol branch of the mevalonate pathway, i., with squalestatin: that leads to large increases in FPP, DolP and CoQ (see Keller RK, BBA 1996, PMID 8399346), and also was found to cause increased Dol chain length (from C95 to C115). Furthermore, Waechter and colleagues found only slight, if any, increase in flux into Dol, CoQ, and prenylated proteins in squalestatin-treated C6 glial cells incubated with 3H-mevalonolactone (see Crick DC et al. J Neurochem. 1995; PMID: 7643114).]  Finally, there are no reports documenting changes in FPP or IPP or CoQ levels in tissues or bodily fluids from RP59 patients or in any extant animal model of the disease.  Given all of the above, we prefer not to go out on a limb at this point to incorporate such speculation as offered by this Reviewer in our review article.

Reviewer 3 Report

Review “Vertebrate Animal Models of RP59: Current status and Future Prospects” analyses and discusses information published in scientific articles (69 cited sources) on retinitis pigmentosa, a rare recessive form of RP. The reviewed animal models are zebrafish model of RP59 and several mouse models of RP59 including K42E Dhdds knock-in mouse, Rod-specific Dhdds knockout mouse, RPE-specific Dhdds knockout mouse, and Nogo-B receptor mutants. Also, a Drosophila model of RP59 is discussed. The authors of the review point out the positive aspects as well limiting factors of the analysed models, doing it very tactfully and ethically. The authors also emphasize that the anatomical structure of the mouse retina is somewhat different from the structure of human retina, thus researches must be aware that this difference may influence gene expression and gene network interactions. Moreover, the authors of the review underline that the described animal models were created relatively recently and are not fully characterized, but despite of that the models have already provided some fundamental insights into the disease. The authors of the review conclude that the described animal models are increasingly used, which gives hope that new information will be added in the near future and the models will offer opportunities for testing new therapeutic interventions to reduce the course of RP59 or, ideally, even prevent the disease.

Dear Authors, there were some small errors in your work: 3.2, row 2 - [55] - is not in bold; row 6 - [6,13,14,56] - is not in bold; 3.3 - 5 lines from the bottom - {60] is not in bold; 3.5 - row 3 - [62] - is not in bold; 4. Discussion - [64] - is not in bold.

Your manuscript is interesting!

Author Response

:  We thank this Reviewer for the overtly positive and complimentary evaluation, and for taking the time and effort to read and review our manuscript.  The points of the critique that require our attention: “Dear Authors, there were some small errors in your work: 3.2, row 2 - [55] - is not in bold; row 6 - [6,13,14,56] - is not in bold; 3.3 - 5 lines from the bottom - {60] is not in bold; 3.5 - row 3 - [62] - is not in bold; 4. Discussion - [64] - is not in bold.”

RESPONSE:   Thank you for catching those minor errors.  We’ve now made those corrections, accordingly.